# Detection of RNA-Dependent RNA Polymerase of Hubei Reo-Like Virus 7 by Next-Generation Sequencing in *Aedes aegypti* and *Culex quinquefasciatus* Mosquitoes from Brazil

**DOI:** 10.3390/v11020147

**Published:** 2019-02-10

**Authors:** Geovani de Oliveira Ribeiro, Fred Julio Costa Monteiro, Marlisson Octavio da S Rego, Edcelha Soares D’Athaide Ribeiro, Daniela Funayama de Castro, Marcos Montani Caseiro, Robson dos Santos Souza Marinho, Shirley Vasconcelos Komninakis, Steven S. Witkin, Xutao Deng, Eric Delwart, Ester Cerdeira Sabino, Antonio Charlys da Costa, Élcio Leal

**Affiliations:** 1Institute f Biological Sciences, Federal University of Pará, Av. Augusto Correa01, CEP 66075-000 Belém, Pará, Brazil; geovanibiotec@gmail.com; 2Laboratório de Vetores, Superintendência de Vigilância em Saúde do Amapá, Rua Tancredo Neves, 1.118, CEP 68905-230 Macapá, AP, Brazil; fredjulio@gmail.com (F.J.C.M.); farmarlisson@hotmail.com (M.O.d.S.R.); edcelhamanu@hotmail.com (E.S.D.R.); 3Lusíada University, Rua Oswaldo Cruz, 179, CEP 11045-101 Santos, SP, Brazil; dani_kstro@hotmail.com (D.F.d.C.); mcaseiro@uol.com.br (M.M.C.); 4Laboratório de Retrovirologia, Universidade Federal de São Paulo, Rua Pedro de Toledo, 781, CEP 04039-032 São Paulo, SP, Brazil; robsonsantos@id.uff.br (R.d.S.S.M.); skomninakis@yahoo.com.br (S.V.K.); 5Faculty of Medicine of ABC, Santo André, SP 09060-870, Brazil; 6Department of Obstetrics and Gynecology, Weill Cornell Medicine, 407 E 61st St, New York, NY 10065, USA; switkin@med.cornell.edu; 7Institute of Tropical Medicine, University of São Paulo, Avenida Dr. Enéas Carvalho de Aguiar, 470-CEP 05403-000 São Paulo, Brazil; sabinoec@gmail.com; 8Vitalant Research Institute, 270 Masonic Avenue, San Francisco, CA 94118-4417, USA; xdeng@vitalant.org (X.D.); eric.delwart@ucsf.edu (E.D.); 9Department of Laboratory Medicine, University of California, Blood Systems Research Institute, 270 Masonic Ave, San Francisco, CA 94118, USA; 10Department of Infectious Diseases, Faculty of Medicine, University of São Paulo, Av. Dr. Arnaldo, 455-Cerqueira César CEP, 01246903 São Paulo, SP, Brazil

**Keywords:** Hubei reo-like virus 7, reovirus, RNA-dependent RNA polymerase, metagenomic, *Aedes aegypti*, *Culex quinquefasciatus*, mosquitoes, insect-viruses, Amazon forest, arbovirus, birds, Brazil

## Abstract

Advancements in next-generation sequencing and bioinformatics have expanded our knowledge of the diversity of viruses (pathogens and non-pathogens) harbored by mosquitoes. Hubei reo-like virus 7 (HRLV 7) was recently detected by the virome analysis of fecal samples from migratory birds in Australia. We now report the detection of RNA-dependent RNA polymerase sequences of HRLV 7 in pools of *Aedes aegypti* and *Culex quinquefasciatus* mosquitoes species from the Brazilian Amazon forest. Phylogenetic inferences indicated that all HRLV 7 strains fall within the same independent clade. In addition, HRLV 7 shared a close ancestral lineage with the *Dinovernavirus* genus of the Reoviridae family. Our findings indicate that HRLV 7 is present in two species of mosquitoes.

## 1. Introduction

Mosquitoes are well recognized arthropod vectors of viral pathogens worldwide [1,2]. In addition, these insects are widely known to harbor a large diversity of seemingly non-pathogenic viruses [3,4,5]. In Brazil, climatic and geographic conditions are very favorable for the proliferation and dissemination of mosquitoes and their viral microbiota: frequent rainfall, year-round high temperature and dense forests. In the last few years Brazil has faced multiple epidemics caused by mosquito-borne viruses such as dengue virus, Zika virus and yellow fever virus [6,7,8,9]. These factors highlight the need for the comprehensive surveillance of virus diversity in mosquitoes. Metagenomic analysis allows for the detection and genetic characterization of novel viruses using high-throughput DNA sequencing technologies [10,11,12,13,14] and has provided new insights into the diversity of RNA viruses infecting invertebrates [15]. Hubei reo-like virus 7 (HRLV 7) is an unclassified RNA virus recently detected by a metagenomic study of invertebrate RNA viruses from insects [12] in China. Nucleotide sequences of HRLV 7 were also identified in fecal samples from Australian migratory birds with a postulated origin from eaten mosquitoes [16].

Here, we report the detection and phylogenetic characterization of HRLV 7 strains from two mosquito species (*Aedes aegypti* and *Culex quinquefasciatus*) in Brazil. Their phylogenetic relationship and detection in different mosquito species increases our knowledge of this recently identified virus.

## 2. Materials and Methods

### 2.1. Mosquitoes Collection

Mosquitoes (Diptera: Culicidae) were collected from the city of Macapá, Amapá state, Northern Brazil (Figure 1) twice a month from January to March 2017. Electric manual aspirators and entomological nets were used to collect the mosquitoes. They were then transported to the laboratory, euthanized with ethyl acetate and morphologically identified using the dichotomous keys of Consoli and Lourenço-de-Oliveira [17]. Up to five females were grouped in pools according to their taxonomic category, place and date of collection. About 105 pools of mosquitoes were stored in a −80 °C freezer. In a similar way, mosquitoes were collected in the city of Santos, São Paulo state, Southeast Brazil (Figure 1), in 2014. About 86 pools of mosquitoes were identified and stored in a −80 °C freezer. Pools of mosquitoes were analyzed according to the following protocol.

### 2.2. Sample Processing and Next-Generation Sequencing (NGS)

The protocol used to perform deep sequencing was described previously by da Costa et al. [18]. Initially, each mosquito pool was homogenized in a 2-mL impact-resistant tube containing lysing matrix C (MP Biomedicals, Santa Ana, CA, USA) added to 900 μL of Hanks’ buffered salt solution (HBSS). The homogenized sample was centrifuged at 12,000 × *g* for 10 min, and approximately 300 μL of the supernatant was then percolated through a 0.45-μm filter (Merck Millipore, Billerica, MA, USA) in order to remove eukaryotic- and bacterial-cell-sized particles. Approximately 100 μL, roughly equivalent to one fourth of the volume of the tube, of cold PEG-it Virus Precipitation Solution (System Biosciences, Palo Alto, CA, USA) was added to the obtained filtrate, and the contents of the tube were gently mixed then incubated at 4 °C for 24 h. After the incubation period, the mixture was centrifuged at 10,000 × *g* for 30 min at 4 °C. Following centrifugation, the supernatant (~350 μL) was discarded. The viral particle enriched pellet was treated with a mix of nuclease enzymes (7 µL of TURBO DNase and 3 µL RNase Cocktail Enzyme Mix-Thermo Fischer Scientific, Waltham, MA, USA; 3 µL Baseline-ZERO DNase-Epicentre, Madison, WI, USA; 3 µL Benzonase-Darmstadt, Darmstadt, Germany; and 3 µL RQ1 RNase-Free DNase and 3 µL DNase A Solution-Promega, Madison, WI, USA) in order to digest unprotected nucleic acids. The resulting mixture was subsequently incubated at 37 °C for 2 h. Viral nucleic acids were then obtained using a ZR and ZR-96 Viral DNA/RNA Kit (Zymo Research, Irvine, CA, USA) according to the manufacturer’s protocol and cDNA synthesis was performed using AMV reverse transcription (Promega, Madison, WI, USA). A second strand of cDNA synthesis was obtained using DNA Polymerase I Large Fragment (Promega, Madison, WI, USA). Then, DNA library analysis was performed using a Nextera XT Sample Preparation Kit (Illumina, CA, USA). The library was deep-sequenced using the HiSeq 2500 Sequencer (Illumina, CA, USA) with 126 bp ends. Bioinformatic analysis was performed according to the protocol previously described by Deng et al. [19]. The singlets and contigs were analyzed via BLAST (BLASTn and BLASTx) to identify similarity to viral sequences in the GenBank database.

### 2.3. Sequence Analysis

Based on the best hits of the BLASTx search, the following sequences of RNA-dependent RNA polymerases (RdRp) from the reference virus genomes of the family Reoviridae, listed by their Genbank numbers, were chosen for phylogenetic analysis: AF291684, AF389452, AF389463, MH085099, MH0850100, MH0850101, DQ087277, KJ191105, KM978416, KM978427, KM978428, KM978429, KP217035, KR704195, KX509952, KX884614, MF161423 and KX884635. Alignment was conducted in Mafft software online [20] and analyzed by the identity matrix tool of BioEdit 7.0.5.3 software [21]. Detailed information of the strains chosen are listed in Appendix A. A maximum likelihood (ML) tree was constructed using PhyML software [22]. Branch support values of the ML tree were assessed using the approximate likelihood ratio test (aLRT). The General Time Reversible (GTR) model and gamma distribution were selected according to the Bayesian information criterion (BIC) implemented in the jModeltest software [23]. The evolutionary history was inferred by using the Maximum Likelihood method and the General Time Reversible model. Estimated GTR frequencies were A:0.3434, G:0.1692, T:0.1988, C:0.2886, and the transition rates were AC:1.3930, AG:2.5688, AT:1.1758, CG:1.3281, CT:2.9047, GT:1.0000. A discrete Gamma distribution was used to model evolutionary rate differences among sites (five categories (+G, parameter = 1.6232)). Selection of the best-fit nucleotide model was performed using the Bayesian Information Criterion (BIC). The lowest BIC scores model that was considered to describe the substitution pattern the best was GTR + gamma distribution (BIC score; 113841.75). The tree with the highest log likelihood (−56589.22) is shown. Initial tree(s) for the heuristic search were obtained automatically by applying Neighbor-Joining algorithms to an instantaneous matrix of pairwise distances estimated using the Maximum Likelihood (ML) approach, and then selecting the topology with the superior log likelihood value. The tree is drawn to scale, with branch lengths measured in the number of substitutions per site. This analysis involved 25 nucleotide sequences. There was a total of 5389 positions in the final dataset.

The Neighbor-Joining tree and the calculation of the mean evolutionary distance within were performed using the MEGA software version X [24]. Bootstrap analysis was performed using 1000 replications. The evolutionary history was inferred using the Neighbor-Joining method. The optimal tree with the sum of branch length = 69.85562761 is shown. The tree is drawn to scale, with branch lengths in the same units as those of the evolutionary distances used to infer the phylogenetic tree. The evolutionary distances were computed using the Jone-Taylor-Thornton (JTT) matrix-based method and are in the units of the number of amino acid substitutions per site. The rate variation among sites was modeled with a gamma distribution (shape parameter = 2.52). The best-fitted model was selected based on BIC scores (BIC score; 156640.583). Estimated amino acid frequencies were: A = 0.077; R = 0.051; N = 0.043; D = 0.051; C = 0.02; Q = 0.041; E = 0.062; G = 0.075; H = 0.023; I = 0.053; L = 0.091; K = 0.06; M = 0.023; F = 0.041; P = 0.051; S = 0.068; T = 0.059; W = 0.014; Y = 0.032; and V = 0.066. This analysis involved 31 amino acid sequences. All ambiguous positions were removed for each sequence pair (pairwise deletion option). There were a total of 1964 positions in the final dataset.

Estimates of average genetic distances over all sequence pairs were performed by the number of base substitutions per site from averaging over all sequence pairs. Standard error estimate(s) were obtained by a bootstrap procedure (500 replicates). Analyses were conducted using the Kimura 2-parameter model. The rate variation among sites was modeled with a gamma distribution (shape parameter = 2.5). All ambiguous positions were removed for each sequence pair (pairwise deletion option). There was a total of 5389 positions in the final dataset.

## 3. Results

We processed a total of 191 pools of mosquitoes from Macapá and Santos that were submitted for NGS. In six pools we detected sequences (3372 nucleotide length on average) that were nearly identical to the RdRp segment of the mosHB235771 Hubei reo-like virus 7 strain (KX884635) (98% and 99% of nucleotide and amino acid identity, respectively). At the amino acid level, the Brazilian HRLV 7 sequences also showed from 39 to 43% identity similarity to *Aedes pseudoscutellaris* reovirus (AAZ94069) and Fako virus (AIW39868), two insect-specific viruses that belong to a recently described genus, *Dinovernavirus* (family Reoviridae). Five of these RdRp sequences of HRLV 7 (named F13, F73, F85, F91 and F99) were detected in Amapá and one in Santos (named S48). The sequences S48 and F13 were detected in *Aedes aegypti* and the sequences F73, F85, F91 and F99 were detected in *Culex quinquefasciatus*. The nucleotide sequences of this study have been deposited in GenBank under the accession numbers MK133923–MK133928.

Pairwise comparison of the RdRp amino acid sequence indicated that HRLV 7 sequences shared almost 40% amino acid identity with *Dinovernavirus* prototype strains, followed by the cypoviruses and oryzaviruses with ~20% and ~18% identity similarity, respectively. We evaluated the amino acid differences in all HRLV 7 strains and found 42 residues out of 1212 of the RdRp. Particularly, we found some residues conserved in Brazilian HRLV 7 that were distinct from the Asian strain HB235771. These differences in the RdRp that may characterize HRLV 7 from South America are: (S180N), (I197V), (T299V), (S809N), (V829L), (K895G), (D898N), (K900E), (T936N), (I987V), (E994D), (N1021T), (S1031G), (S1037V), (H1210Y).

A Maximum likelihood (ML) phylogeny based on RdRp sequences of representative members of the *Cypovirus* and *Dinovernavirus* genera and nucleotide sequences of HRLV 7 showed that all sequences of HRLV 7 are monophyletic; they form a unique clade with high branch support (Figure 2b). Besides, the HRLV 7 clade shares a common ancestry with the genus *Dinovernavirus*. To better illustrate the genetic diversity of HRLV 7 strains, we estimated evolutionary distances between sequences (pairwise distances) by computing the proportion of nucleotide differences between each pair of sequences. HRLV 7 and *Dinovernavirus* strains had 47% genetic distance and the genetic distance of all HRLV 7 RdRp sequences was only 2% (Figure 2b), which suggests that they belong to a single replicating lineage. Futhermore, the genetic distances between the *Dinovernavirus* clade plus the HRLV 7 clade and the *Cypovirus* clade was 60%.

## 4. Discussion

RdRp is the most conserved protein among reoviruses and is often used for phylogenetic analysis within this family. It has been suggested, as a rule, that a percentage of amino acid identity similarity greater than 30% of the RdRp region is needed for classification within a genus of the Reoviridae family [25]. Our identity analysis and phylogenetic tree indicated that HRLV 7 is monophyletic and is related to the *Dinovernavirus* genus (Figure 2b). The genus *Dinovernavirus* includes the only reoviruses known that contain a nine-segment double-strand RNA genome. Other criteria are required for the classification of genera within the *Reoviridae* family, such as sequences of conserved terminal nucleotide motifs and structural genes [26] that need full-length genomes for detection. Therefore, more phenotypic and genomic studies are necessary to confirm the taxonomic classification of HRLV 7.

HRLV 7 was previously detected from a mixture of mosquito species comprised of *Aedes* sp., *Armigeres subalbatus*, *Anopheles sinensis*, *C. quinquefasciatus* and *C. tritaeniorhynchus* [12]. In our study, nucleotide sequences of HRLV 7 strains were detected in one pool of *A. aegypti* and five pools of *C. quinquefasciatus*¸ indicating that both mosquito species can be infected by HRLV 7. The detection of a single HRLV 7 lineage from multiple species, rather than a lineage adapted to each specific species, suggests that horizontal transmission between species may be occurring extensively, possibly through the infection of and feeding on common sources [27]. Moreover, *A. aegypti* and *C. quinquefasciatus* are distributed worldwide, predominantly in tropical regions, suggesting that HRLV 7 may potentially be present in other tropical regions of the world. In this study, HRLV 7 was detected in Santos and Macapá, two regions that are nearly 2700 km apart. In addition, HRLV 7 has previously been detected in China and Australia [11]. Our evolutionary analysis showed that HRLV 7 strains have low genetic distance (~2%) in their RdRp region [3,16]. RNA viruses are more prone to mutations, due to low fidelity transcription of RNA polymerase [28]. Therefore, finding similarity values of nearly 100% in geographically distant HRLV 7 strains suggest that this virus is dispersed over long distances.

In summary, viruses present in mosquitoes can travel long distances and enter new countries or even continents, either by the unintentional aid of human transportation, such as aircraft, trains and vehicles, or by birds that feed on infected mosquitoes, travel long distances, and then deposit these viruses in their feces, as reported by Vibin et al. [16]. Determining the host range of viruses is essential to understanding the process of potential transmission. Further surveillance and molecular studies are required to fully understand the ecology and evolution of HRLV 7.

## Figures and Tables

**Figure 1 viruses-11-00147-f001:**
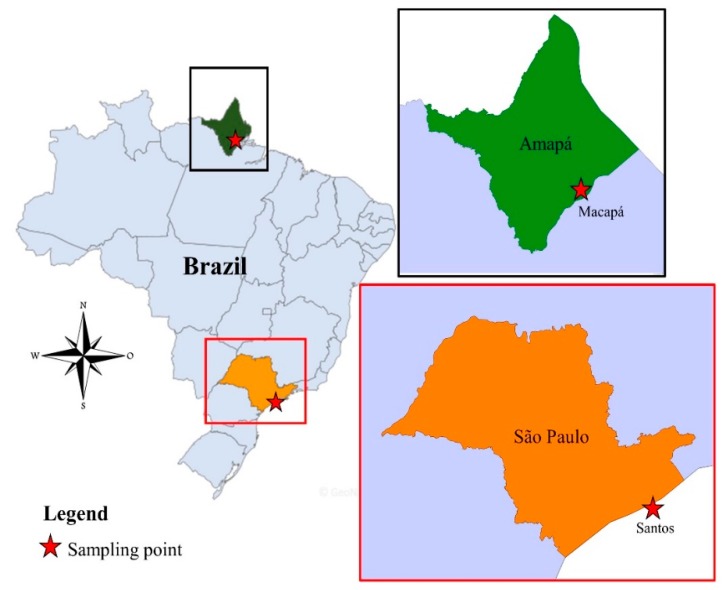
Location of the study area. From left to right: map of Brazil highlighting the state of Amapá and São Paulo, map of Amapá (Up) and São Paulo (down) showing the cities of Macapá and Santos, respectively.

**Figure 2 viruses-11-00147-f002:**
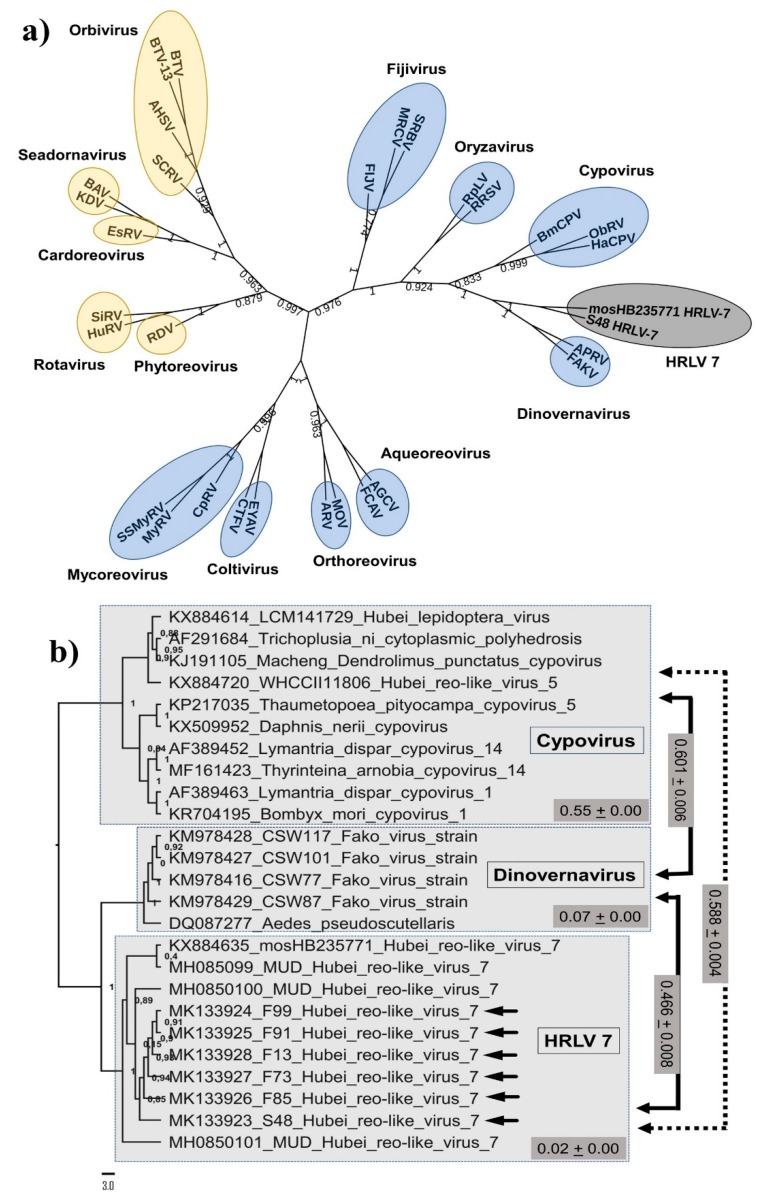
Phylogenetic trees constructed with the *RdRp* gene region of reoviruses. (**a**) Phylogeny of representative reovirus sequences based on the alignment of full-length protein sequences of RdRp. HRLV 7 strains are highlighted in the gray circle. Members of the subfamily Spinareovirinae are highlighted in blue, and members of the subfamily Sedoreovirinae are highlighted in yellow. The various genera are also indicated in bold. Tip labels include the International Committee on Taxonomy of Viruses (ICTV) abbreviations. This tree was constructed using the Neighbor-Joining method and assuming a JTT model [24]. Values on the node of the trees indicate the statistical support based on a bootstrap test using 1000 replicates. Sequences used for phylogenetic analyses are shown in Appendix A. (**b**) Maximum likelihood (ML) tree based on nearly full-length of nucleotide sequences of RNA-dependent RNA polymerases (RdRp) shows three groups (gray areas) containing strains from *Dinovernavirus*, *Cypovirus* and HRLV 7 strains. The Brazilian strains described in this work are indicated by arrows. The filled gray rectangle indicates genetic distance within and between the clades. The ML tree was inferred using the General Time Reversible (GTR) + gamma distribution model and values on node of the trees indicate the statistical support based on the approximate likelihood ratio test (aLRT).

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
