# Peer review of "Detection of RNA-Dependent RNA Polymerase of Hubei Reo-Like Virus 7 by Next-Generation Sequencing in Aedes aegypti and Culex quinquefasciatus Mosquitoes from Brazil"

_viruses, 2019, doi:10.3390/v11020147_

Reviewer 1 Report

The authors have addressed all of the changes that needed to be made for publication.

Author Response

We deeply appreciated the detailed and insightful comments made by this referee

Reviewer 2 Report

Major comments:

Whilst effort has been made to address previous reviewer’s comments, and the manuscript has been improved, issues remain. Some important comments were not adequately addressed, and in some cases where changes were made, these have created additional issues. For example, the point raised by the other reviewer that the sequences may originate from a transcribed and integrated element of the mosquito genome has not been properly discussed. All these issues must be adequately addressed for the manuscript to be acceptable for publication.

Minor comments:

1.      Line 39. I believe the reference to Aedes aeggenbankypti is a typographical error. Please correct.

2.      Line 39-40. The part of the sentence containing “different species of mosquitoes” should be changed to “two species of mosquitoes.”

3.      Line 101. “GenBank’s Virus” does not mean anything. Please correct.

4.      Lines 102-115. As requested in the original review, all parameter settings for software should be given.

5.      Lines 140-142. As stated previously, the genetic distance and the proportion of nucleotide identity (or difference) are not exactly the same thing. In the text it says “we estimated evolutionary distances between sequences (pairwise distances) by computing the proportion of nucleotide differences between each pair of sequences.” In responding to comments, the authors make reference to [Tajima F & Nei M (1984) Estimation of evolutionary distance between nucleotide sequences. Molecular Biology and Evolution1:269-285]. They also say they used MEGA to calculate genetic distances but the method used (e.g. p-distance or JTT etc.) is not stated. Please clarify, stating the method used to calculate the distance in the manuscript.

6.      Lines 175. As an isolate was not obtained, please replace “isolation” with “detection.”

7.      Lines 187-192. “Another important feature of our study is that only the RdRpol of HRLV 7 was detected. Interestingly, in the same pools that HRVL 7 was detected we also found other viruses, including segmented viruses such as Fako virus (data not shown). Likewise, prior investigations also reported only the RdRp of HRLV 7 [3, 16]. Thus, the absence of other genomic regions of HRLV 7 does not seem to be a technical limitation because distinct approaches were used herein and in previous studies.” I cannot understand the points the authors are trying to make here. Please clarify or perhaps the paragraph should be removed.

8.      Line 187. In some cases RdRp is used for RNA-dependent RNA polymerase, and in another RdRpol; please use one for consistency.

9.      Line 193. Please replace “In sum” with “In summary.”

10.   Figure 2. In the revised version received there appears to be two versions of Figure 2, and the lower one appears to be a more recent version. One will need to be removed.

11.   The author’s comment in the response “Our point is that RNA viruses are prone to mutations and Culex and Aedes are separated over 52–54 million years (Arensburger et al. Science. 2010 Oct 1;330(6000):86-8. doi: 10.1126/science.1191864) then we should expect more than 2% of diversity in the RdRp of HRLV 7” suggests they believe the vector and the virus are co-evolving. In saying this, they seem to have neglected the possibility that virus could be exchanged between mosquito species by feeding on a common host. However, they state that this is a possibility in the manuscript so there seems to be some confusion here. This brings me back to my original request, for the authors to consider the rate of change for the RdRp region based on observations in the literature, if possible, to support their assertion that the divergence is low.

Author Response

We deeply appreciated the detailed and insightful comments made by this referee

This manuscript is a resubmission of an earlier submission. The following is a list of the peer review reports and author responses from that submission.

Round  1

Reviewer 1 Report

In the manuscript ‘Novel reovirus in Aedes aegypti and Culex quinquefasciatus mosquitoes from Brazil closely related to virus in bird feces.’ the authors report the detection of the RdRP gene of Hubei reo-like virus 7 in mosquito pools from Brazil. The authors further classify this virus as a Dinovernavirus and characterize the phylogeny of this RdRP gene. The manuscript represents only a small addition to current research – the ‘virus’ which the researchers characterize was really only one of the genes (RdRP) and it is unclear why the full length sequence of this virus was not identified and described (unless it is only a viral element). Based on the methods, the authors clearly attempted to only sequence packaged viral RNA/DNA in this study. The library preparation should thus have resulted in mostly pure viral reads after sequencing. Obtaining and including genome sequences encoding structural proteins in the presented analysis would have added to the impact of the manuscript. A further major comment is regarding the use of the term ‘novel virus’ – the authors claim in the title, abstract and other parts of the manuscript that they discovered this novel virus. However, the authors also show close sequence similarity to the Hubei reo-like virus 7 found in a previous study from China (and Australia). It is thus not the detection of a novel virus, rather than the detection of a previously identified virus in Brazilian mosquitoes. The wording should be adjusted. Yet, the manuscript is for the most part clearly written/presented, and there are no obvious major concerns with the experimental design, except that further experiments could have provided valuable information and increased the impact of the work. Aside from the overall lack of depth of the study, I just have a few minor comments for the authors to address. Title: the title could be improved (it may read better as a complete sentence, and the word novel should be removed) Line 84: ‘The pellet rich in viral particles …’ – how do the authors know that the pellet was ‘rich in viral particles’? Was there some sort of control used? I am mainly critical of these steps because of the reporting of a partial genome (RdRP sequence) that may not represent sequences obtained from viral particles, but rather sequences present in the genome/transcriptome of the collected mosquitoes. Lines 84-85: the authors refer to ‘a mix of nuclease enzymes’. This is not appropriate in detail for this part of the methods. The authors should provide information on which nucleases and what concentrations were used, so that the reader can evaluate whether these nucleases would appropriately digest free RNA/DNA. Lines 124-126: the authors use the words ‘isolates’. While the authors likely enriched for viral particles using the methods described in lines 77-85, there is no positive control or confirmation by EM or similar method showing that only virus particles were obtained. It may be more appropriate to refer to sequences here. Virus isolates should only be used if virus was clearly isolated from the mosquito pools, for example by growing up virus on cell culture or possibly by showing viral particles using EM (both methods would have been useful additions to the study). Raw sequencing data should be made available prior to publication (SRA data base or similar). Line 163: ‘Others’ should be ‘Other’ Lines 165-168: Here the authors bring up the point that no full length genome of HRLV 7 is available. The authors should also discuss why it was not detected in their study and whether HRLV 7 is a true ‘virus’ or a possibly a viral element integrated in and transcribed from the mosquito genome. An RNA gel of the ‘isolated’ virus prepared here by PEG-purification for example may have been able to show whether 9 RNA segments are present. A PCR on DNA isolated from these mosquito pools may have helped identify whether this is an integrated viral sequence or not. Since none of these experiments were performed, they should at least be discussed.

Reviewer 2 Report

Ribeiro2018 comments

Major comment:

This manuscript is a description of the detection of Hubei reo-like virus 7 sequences in pooled mosquito material. Whilst interesting, the manuscript lacks critical detail in methods and description of results. This must be corrected to be suitable for publication. Importantly, the claim that Hubei reo-like virus 7 is in the genus Dinovernavirus has not been justified. Also, there is inconsistent use of terms such as identity, similarity, divergence and genetic distance. The authors must modify the text ensuring that they are using such terms correctly and consistently.

Minor comments:

1.      Title. The title of the manuscript states this is a “novel” virus, however, there are at least two other papers cited referring to this virus. Hence, the virus has been recently discovered but is not novel, hence “novel” must be removed from the title. In fact, I suggest a different title better capturing the essence of the paper may be appropriate.

2.      Line 32. Please change “has” to “have.”

3.      Line 34. I think the authors mean “detected” rather than “discovered.” Please correct.

4.      Line 36. Please replace “species” with “mosquito species.”

5.      Line 39. By “highly disseminated” do the authors mean in multiple species of mosquitoes? If so, this seems inappropriate as there were only pools from two different species of mosquitoes. If they mean geographically then this point is out of context. Please correct.

6.      Line 47. Do mosquitoes migrate? Perhaps “spread” is a better word.

7.      Line 81. Numerals should not start a sentence (i.e. 100 ul). Please correct.

8.      Lines 94-114. The “Evolutionary Analysis” section reads poorly and should be amended to improved clarity.

9.      Line 101. What does “Based on best result analysis described above…” mean? Please clarify.

10.   Line 116. Please change “submitted to NGS” to “submitted for NGS.”

11.   Line 123. Please change “searches” to “search.”

12.   Lines 127-128. Please change “sequence” and “number” to plural.

13.   In general, there is only very minimal description of methods. For example, it is unclear how the authors performed their sequence searches. For example, they say blastn and blastx were done, but the database used was not stated (i.e. GenBank’s Virus?).

14.   Lines 116-120. When the authors say 5 sequences of Hubei reo-like virus 7 strain were obtained from their pools, what does this mean exactly? Does it mean that one sequence was obtained separately in 5 out of 191 pools? Were these isolates, as referred to later in the text? What was used as a reference to obtain these sequences, or was de novo assembly used? If a reference sequence was used the GenBank accession number must be stated.

15.   Lines 124-127. Related to the comment directly above, the authors now mention “isolates.” Do the authors mean an isolate was obtained by tissue culture as there is no mention of how these isolates were obtained? If isolates were obtained, this should be stated and the methods given in the results.

16.   129-130. It is claimed that Hubei reo-like virus 7 group is in the genus Dinovernavirus. However, Hubei reo-like virus 7 groups in a separate sister clade to the dinovernaviruses. The authors need to calculate genetic distances between these groups and conclude whether Hubei reo-like virus 7 is a member of an existing, or possibly novel, genus.

17.   Lines 134-135. This sentence does not make sense and needs to be corrected.

18.   Lines 145-153. The region of RdRp used in the analysis, including amino acid position and any relevant domains included, must be stated. In addition, why was neighbour joining used for the tree in Figure 2A and maximum likelihood used for Figure 2B? Maximum likelihood is generally considered a superior method to neighbour joining. The use of each method must be justified.

19.   Line 146. “(bootstrapping (up) and aLRT (down))” where is this on the figure?

20.   Line 147. “Reovirus” should not be capitalized here. Please correct.

21.   Line 163. Please change “others” to “other.”

22.   Lines 171. Please change “five pool” to “five pools.”

23.   Lines 180-181. Without reference to the rate of change for RdRp in similar viruses it is hard to determine whether 2% divergence is low as suggested. Please refer to the literature if possible.

24.   Line 184. I would suggest changing “easily” to “relatively rapidly.”

25.   The parameter settings for the bioinformatics analyses conducted in the paper were not given. All such parameters must be given.

26.   Figure 2 and lines 140-142. The authors refer to genetic distances. How were these calculated (i.e. MEGA program), or are they using differences in amino acid identity scores? Correct genetic distances should be calculated and used to justify taxonomic determinations in reference to the literature with this virus group. As stated above, this should be the basis of whether this virus is a member of the dinovernaviruses or possibly even a new genus.

27.   Supplementary material. A link to supplementary file returns no information.